# MicroRNA and Oxidative Stress Interplay in the Context of Breast Cancer Pathogenesis

**DOI:** 10.3390/ijms20205143

**Published:** 2019-10-17

**Authors:** Giulia Cosentino, Ilaria Plantamura, Alessandra Cataldo, Marilena V. Iorio

**Affiliations:** 1Molecular Targeting Unit, Research Department, Fondazione IRCCS Istituto Nazionale dei Tumori, 20133 Milan, Italy; giulia.cosentino@istitutotumori.mi.it (G.C.); ilaria.plantamura@istitutotumori.mi.it (I.P.); 2IFOM Istituto FIRC di Oncologia Molecolare, 20139 Milan, Italy

**Keywords:** oxidative stress, miRNAs, breast cancer, ROS

## Abstract

Oxidative stress is a pathological condition determined by a disturbance in reactive oxygen species (ROS) homeostasis. Depending on the entity of the perturbation, normal cells can either restore equilibrium or activate pathways of cell death. On the contrary, cancer cells exploit this phenomenon to sustain a proliferative and aggressive phenotype. In fact, ROS overproduction or their reduced disposal influence all hallmarks of cancer, from genome instability to cell metabolism, angiogenesis, invasion and metastasis. A persistent state of oxidative stress can even initiate tumorigenesis. MicroRNAs (miRNAs) are small non coding RNAs with regulatory functions, which expression has been extensively proven to be dysregulated in cancer. Intuitively, miRNA transcription and biogenesis are affected by the oxidative status of the cell and, in some instances, they participate in defining it. Indeed, it is widely reported the role of miRNAs in regulating numerous factors involved in the ROS signaling pathways. Given that miRNA function and modulation relies on cell type or tumor, in order to delineate a clearer and more exhaustive picture, in this review we present a comprehensive overview of the literature concerning how miRNAs and ROS signaling interplay affects breast cancer progression.

## 1. Oxidative Stress

Reactive oxygen species (ROS) are oxygen-derived small molecules in the form of free radicals (i.e., contains one or more unpaired electrons) or non-radicals [1]. Among the most biologically relevant species there are the superoxide anion radical (O_2_^−•^), the hydroxyl radical (OH·) and hydrogen peroxide (H_2_O_2_). At first, it was thought that these molecules were only metabolic waste, deleterious for nucleic acids, lipids and proteins; scientists, however, discovered that ROS are used by the cell as messages to activate different physiological signaling cascades [2,3]. In fact, in a biological system, the balance between the concentration of ROS and the activation of antioxidant mechanisms is finely tuned [4]. When this equilibrium lacks, the phenomenon of oxidative stress occurs, causing the alteration of intracellular molecules, such as DNA and RNA. A shift towards ROS production, thus, triggers a wide range of cellular responses, even apoptosis or phagocytosis, depending on the amplitude of the shift. Several endogenous and exogenous sources can trigger ROS production. In response to stimuli like cytokines and growth factors, NADPH oxidases (NOXs) and mitochondria produce the larger percentage of ROS. NOXs and metabolic complexes I, II and III present on the mitochondrial inner membrane generate, for example, the radical superoxide starting from a molecule of oxygen. Dangerous levels of ROS can be reached also after prolonged exposure to radiations and carcinogens, along with DNA damaging drugs. The major mutagenic product of DNA oxidation is 8-hydroxyl-2′-deoxyguanosine (8-OHdG). Figure 1 summarizes principal sources producing ROS and main regulators and pathways influenced by ROS production (Figure 1). Cancer cells are usually in a chronic state of oxidative stress, which they are able to exploit to sustain a proliferative and aggressive phenotype. Moreover, due to their detrimental action, ROS can also initiate tumorigenesis [5]. It is thus important not to overlook the impact of such phenomenon on every cellular process and, in particular, on those crucial for the development and progression of a neoplastic disease.

## 2. Breast Cancer

Breast cancer is the second most commonly diagnosed cancer worldwide and the leading cause of cancer death in women [6]. The severity and aggressiveness of breast cancer is evaluated by examining physical and anatomical properties of the disease, in particular by using histological grading and TNM staging, where T (0–4) is used to describe the size and location of the tumor, N (0–3) accounts for the lymph node invasion and M measures the spread of the tumor as distant metastasis [7].

The therapeutic regimen is finally driven by the characterization of the breast cancer subtype according to the immunohistochemical evaluation of three markers: Estrogen receptor (ER), Progesteron Receptor (Pgr) and HER2 (Epidermal Growth Factor Receptor 2) [8]. Tumors lacking the expression of these three markers are called triple negative breast cancers (TNBCs). A major contribution to the increase in survival rate has been provided by the improvement in the therapeutic regimens as well as in early diagnosis. Moreover, it is fundamental to develop always more personalized drugs for different cancers subtypes.

The advent of the genomic era disclosed the complexity of breast cancer. For the first time, in 2000, Perou and colleagues classified the disease in five specific subtypes according to intrinsic gene expression: Luminal-A, Luminal-B, HER2-positive, Basal-like and Normal-like [9]. Further studies later identified a new subtype, the so called claudin-low [10], which accounts for 7–14% of all breast cancers. Moreover, one of the most important applications of the breast cancer molecular classification lies in its ability to identify groups with a different outcome and response to treatments [11,12,13,14,15].

## 3. Oxidative Stress and Breast Cancer

Breast cancers, in particular estrogen receptor-positive malignancies, are characterized by significant high levels of 8-OHdG, and their detection in blood serum is reported to have prognostic value [16,17,18].

Estrogen is a major driver of mitochondrial ROS production. It activates redox-sensitive proteins involved in cell proliferation and anti-apoptotic pathways. In order to sustain such signaling without risking cell cycle arrest and apoptosis, estrogen enhances also an antioxidant response by inducing, for example, the transcription factor Nuclear-erythroid-2-related factor 2 (NRF2). This enzyme is the main redox master regulator; under oxidative stress, its inhibitor Kelch-like ECH-associated protein 1 (Keap1) undergoes a conformational change that allows NRF2 dissociation and consequent translocation to the nucleus, where it enhances the transcription of different ROS-counteracting agents [19,20]. Numerous evidence shows that NRF2 is overexpressed in breast cancer, where it promotes cell survival, proliferation, migration and metastasis [21,22,23].

Additionally, it is important to note the interplay between NRF2 and BRCA1. Gorrini C. et al. demonstrated that BRCA1 enhances and stabilizes NRF2 expression and that estrogen is able to partially mimic this action in BRCA1-null cells [24,25].

Moreover, in 2014, Victorino V. J. et al. analyzed the effect of HER2 overexpression on the oxidative systemic profile in breast cancer patients [26]. The results showed that HER2-overexpressing malignancies are characterized by an enhanced oxidative stress, attenuated by increased SOD and stabilized gluthatione (GSH) levels, which are indicative of an active antioxidant response. In the same year, Kang H. J. et al. reported that also HER2 interacts with NRF2 to promote the transcription of antioxidant and detoxification genes and that this partnership confers drug resistance to human breast cancer cells [27]. Antioxidants can, thus, favor breast neoplastic transformation: by reducing ROS concentrations they can prevent ROS-dependent cell death [28,29]. Therefore, the role of antioxidants in breast cancer is often controversial; for example antioxidant superoxide dismutase 2 (SOD2), which converts the highly toxic radical superoxide into more stable hydrogen peroxide in the mitochondria, can act both as an oncogene and as a tumor suppressor. In fact, it is found downmodulated in early-stage breast cancer while upregulated in advanced tumors [30,31]. Despite these results, SOD mimics have been proposed for therapeutic purposes [32,33]. Catalase, glutathione peroxidases, and peroxiredoxins are among the other antioxidant enzymes which balance ROS production. In 2017, Bao B. et al. demonstrated that the addition of a re-engineered protein form of the catalase enzyme to EGFR-inhibitor erlotinib treatment helps overcoming resistance by specifically targeting the stem-like portion of TNBC cells [34]. Conversely, peroxiredoxin-1 (PRDX1) downmodulation was shown to be beneficial for breast cancer therapy, especially in concomitance with prooxidant agents [35]. Moreover, specific acquaporins allow H_2_O_2_ to cross cell membranes more rapidly than by sole diffusion [36]. In breast cancer, Aquaporin-3 has been proposed as target for therapy due to its role in CXCL12/CXCR4-dependent cancer cell migration [37].

Finally, damages to RNA molecules have to be considered equally harmful [38]. For example, it is of particular relevance for this review the impact on miRNA biology and the consequent influence on different regulation networks [39].

## 4. MicroRNAs

MicroRNAs (miRNAs) are small single strand molecules (~18–25 nucleotides), they are non-coding RNAs that are able to control gene expression at post-transcriptional level [40]. MiRNA biogenesis starts when RNA polymerase II/III transcribes for a long primary transcript with a hairpin structure, called pri-miRNAs [41]. The pri-miRNA is the substrate of Drosha and Dicer, two members of the RNase III family enzymes. First, Drosha cleaves the pri-miRNA in a ~70-nucleotide pre-miRNA into the nucleus, which is then exported into the cytoplasm by the Exportin-5 Ran-GTPase, where Dicer catalyzes its conversion to a short miRNA/miRNA* duplex (~20 bp). To complete the miRNA biogenesis, the transactivation-responsive RNA-binding protein (TRBP) leads to the assembling of the miRNA-induced silencing complex (miRISC), mediating the interaction between DICER and Argonaute protein (AGO1, AGO2, AGO3 or AGO4). Finally, the miRISC complex selects one single strand of the duplex (mature miRNA), which recognizes the “seed” region on the target mRNA, usually placed at the 5′ UTR, inducing translational repression or deadenylation and degradation. The small RNA lin-4 was the first no-coding RNA discovered in *Caenorhabditis elegans*, involved in the larval development [42]. Afterwards, several studies have pointed out the importance of these small molecules; currently it is well known that miRNAs are involved in almost every biological process in mammals, including oxidative stress and cancer [43]. Indeed, miRNAs can act as oncosuppressors or oncogenes, which are generally found respectively downregulated and upregulated in tumor cells (e.g., miR-205 and miR-21, respectively). In 2005, Iorio M.V. et al. discovered a panel of dysregulated microRNAs in breast cancer: miR-10b, miR-125b, and miR-145 were down-regulated, and miR-21 and miR-155 were up-regulated, suggesting that they could have a role in breast cancer disease [44].

Recently, we reported that miRNAs have a relevant role in DNA damage response, occurring following an exogenous oxidative stress, such as chemotherapy [45,46]. In fact, miRNAs have the capability to target several genes involved in the DNA repair machinery, regulating therapy responsiveness. Here, we review the literature concerning the role of miRNAs in the regulation of the major actors and principal pathways altered by oxidative stress in breast cancer.

## 5. MiRNAs Modulate Oxidative Stress Master Regulators: NRF2 and NF-κB

NRF2 is an important transcription factor which induction, or derepression, depends on the redox status of the cell. Normally, NRF2 is found inactive in the cytoplasm bound to its homodimeric repressor Keap1, which anchors the protein Cullin-3 (CUL3) to form an E3 ubiquitin ligase complex; the complex is responsible for NRF2 ubiquitination and consequent proteasomal degradation [47]. When cellular ROS concentrations increase, specific Keap1 cystenyl residues are modified and NRF2 is released and free to translocate into the nucleus, where it recognizes the so called “Antioxidant Responsive Elements (ARE)” sequences on target gene promoters and enhances the transcription process [48]. NRF2 promotes the expression of antioxidants and detoxifying enzymes and, initially, it was thought to act as a defensive agent against tumorigenesis. However, as previously explained for SOD2, an excessive reduction of ROS levels can prove counterproductive. Therefore, it is not unusual to find contradictory literature concerning the prospective of using NRF2 inhibitors for therapeutic purposes [49,50]. NRF2 pathogenic activation and accumulation can be triggered by different events; one of the most frequent alterations concerns Keap1 expression or its ability to stably bind and degrade NRF2 [51]. MiRNAs were found to exert this oncogenic activity, NRF2 induction, in different malignancies [52,53,54]. In 2011, Eades G. et al. demonstrated for the first time a miRNA-dependent Keap1 regulation in breast cancer: miR-200a targets Keap1 mRNA and induces its degradation [55]. Interestingly, the same group published the same year an additional paper describing NRF2 inhibition by miR-28 in MCF7 breast cancer cell line [56]. Two other miRNAs, miR-93 and miR-153, have been reported to target NRF2 and their overexpression is associated with breast carcinogenesis [57,58]. This evidence validates once more the context-specific value of NRF2 modulation (Figure 2A).

The same concept can be translated to the other redox master regulator, the nuclear factor-kB (NF-κB). NF-κB can be found as both homo- and heterodimer of five distinct proteins, RelA, RelB, c-Rel, p50 and p52. It is inhibited in the cytoplasm by the IκB families, which interfere with the target activity by interacting with its important Rel homology domain (RHD), implicated in the formation of dimers and DNA binding [59].

IκB proteins are generally degraded in response to inflammatory cues like TNFα and lipopolysaccharide (LPS). The consequent NF-κB signaling, modulated by ROS, is cell type and context specific. This is probably due to the transcription factor wide range of action: cell growth, proliferation, migration and apoptosis are among the pathways it influences [60,61]. NF-κB signaling is frequently found dysregulated in human cancers [62]. In breast cancer, the protein is reported as constitutively activated and associated to aggressive and chemoresistant malignances [63,64,65]. MiRNAs play an important part also in this scenario. First of all, NF-κB favors breast cancer cell invasion by inducing the expression of the oncomiR miR-21 in response to DNA damage [66]. Wiemann S. and his group, instead, published different papers on NF-κB-regulating miRNAs over the years [67,68,69,70]. In 2012, they demonstrated the tumor suppressive role of miR-520/373 family in ER-negative breast cancer, through the targeting of NF-κB and TGF-β signaling pathways. In the same context, in 2013, miR-31 was seen to sensitize cancer cells to apoptosis by impairing NF-κB pathway. In 2015, miR-30c-2-3p was shown to reduce proliferation and invasion of MDA-MB-231 cells through the downmodulation of TNFR/NF-κB signaling and cell cycle proteins. Conversely, in 2017, a role as an oncomiR was attributed to miR-1246, which was reported to induce the NF-κB pro-inflammatory signaling in breast cancer cells. The same year, another group discovered that miR-221/222 promote stem-like properties and tumor growth of breast cancer via targeting PTEN and sustained Akt/NF-κB/COX-2 activation (Figure 2B) [71].

Despite having an oscillatory expression in a physiological context, NF-κB thus emerges from the literature presented as a proper oncogene in breast cancer. Moreover, due to its broad spectrum of interactions, numerous are the miRNAs involved in the regulation of the signaling cascade and, consequently, many are the hints for therapeutic interventions.

## 6. MiRNAs Modulate Pathways Altered by Oxidative Stress

### 6.1. Metabolism

The main goal of cancer cells is proliferation and survival. Such activities require a great amount of energy in a short period of time. Therefore, cancer cells tend to modify their metabolism in order to respond to this demand. According to the known Warburg effect, cancers prefer a rapid glycolysis to the more efficient mitochondrial oxidative phosphorylation. This switch also allows avoiding an excessive mitochondria-related production of ROS. Interestingly, it has been suggested that the latter could be the primary reason for the metabolic reprogramming [72]. In breast cancers, the metabolic status seems to be linked to the molecular subtype. In fact, the more aggressive TNBCs are characterized by a glycolytic phenotype, while luminal malignancies retain oxidative phosphorylation as the major source of energy [73]. It is important to note that it is not unusual to find heterogeneity also among cells of the same tumor mass, a scenario that can be as deleterious as a predominant Warburg setting. In 2018, our group indeed proposed that, starting from a mixed population of TNBC cells, pushing all the cells towards a glycolytic phenotype could become counterproductive for the tumor. Through the downmodulation of the lactate transporter MCT1, miR-342-3p is able to disrupt the energetic fluxes between neighboring glycolytic and oxidative cells, promoting the shift and ultimately triggering a competition for glucose [74]. It has been demonstrated that glucose deprivation induces oxidative stress in cancer cells [75]. One of the most cited mechanisms of breast cancer cell metabolic reprogramming that involves miRNAs is miR-155 promotion of hexokinase II (HKII) expression, necessary to start glycolysis. This miRNA modulates multiple pathways that control HKII: first, miR-155, through the direct downmodulation of C/EBPβ, reduces miR-143, a HKII inhibitor; second, the miRNA frees STAT3 from its suppressor SOCS1 to enhance HKII; third, miR-155 positively regulates HKII by interfering with the PIK3R1-FOXO3a-cMYC axis [76,77,78]. Another recent example is the work by Eastlack S. C. et al. that demonstrated miR-27b promotes breast cancer progression by targeting Pyruvate Dehydrogenase Protein X (PDHX), thus altering cell’s metabolic configuration [79]. PI3K/Akt pathway, which players are frequently mutated in breast cancers, deeply impacts on metabolism and ROS production by directly regulating mitochondrial bioenergetics and NOX enzymes. Vice versa, oxidative stress activates PI3K and suppresses the activity of PTEN, inhibitor of PI3K/Akt signaling [80,81,82]. Due to the relevance of the pathway, numerous are the miRNAs found implicated in its regulation in breast cancer. Among the latest reported, there are the tumor suppressor miR-204-5p, which targets PIK3CB, and the PTEN-inhibiting oncomiRs miR-1297 and miR-498 (Figure 3A) [83,84,85].

### 6.2. Hypoxia

Hypoxia refers to a pathological level of oxygen tension, caused by the high proliferative rates of cancer cells and insufficient vasculature. The lack of oxygen supply, thus, induces cancer cells to undergo epithelial-to-mesenchymal transition (EMT), which corresponds to the acquisition of migratory and invasive properties, stem-like features and resistance to apoptosis. The main player in this context is the transcription factor HIF-1α, which stimulates angiogenesis and triggers a positive feedback loop on proliferation pathways. As a consequence, oxidative stress increases upon re-oxygenation and mitochondrial electron leaks. Numerous studies showed a link between the miRNA’s role and the hypoxia in the breast cancer initiation and progression. The first miRNA to be pointed out is miR-210. In 2007, this miRNA emerged as part of the miRNA signature of hypoxia and the year after it was elected as an independent prognostic factor in breast cancer [86,87]. Moreover, Liang H. and colleagues investigated miR-153 mechanism of action in breast cancer; showing that this miRNA acts as a tumor suppressor by targeting HIF-1α [88]. In fact, miR-153 inhibits migration, proliferation and tube formation in HUVEC cells and angiogenesis in MDA-MB-231 in vivo model through the inhibition of the HIF-1α/VEGFA axis. In another paper, the high expression of miR-191 in breast cancer cell lines induces a more aggressive tumor under hypoxia [89]. Consequently, the authors suggest that miR-191 inhibition may be exploited as a new therapeutic option for hypoxic breast cancer. In addition, miR-18a targets HIF-1α, which high expression is associated with shorter DMFS (distant metastasis-free survival) in patients with basal-like breast tumors [90]. In metastatic MDA-MB-231 cells, ectopic miR-18a expression reduces both primary tumor and lung metastasis. Another miRNA reported targeting HIF-1α is miR-497, thus, it represses the hypoxic conditions and for this reason it is usually downregulated in breast cancer cells [91]. MiR-497 also targets a pro-angiogenic molecule, VEGF (vascular endothelial growth factor) and its ectopic expression reduces tumor growth and angiogenesis in breast cancer tumor model (Figure 3B). In conclusion, we could support the strategic role of miRNAs in the tumor progression and in particular in hypoxia and metastasis and we could speculate the possibility to use miRNAs as therapeutic tools to reduce tumor aggressiveness and dissemination.

### 6.3. Response To Therapy

It is well known that miRNAs can influence response to therapy in breast cancer. Moreover, they are under investigation as potential therapeutic tools, alone or in combination with standard therapy to impair cancer progression. Chemotherapy and radiotherapy still represent the standard therapy for breast cancer; miRNAs are able to target different genes reducing drug resistance and promoting therapeutic response. Indeed, in 2016, we reported that miR-302b, by targeting E2F1 and DNA repair, enhances cisplatin response in breast cancer cells [45]. Chemotherapy drugs, such as platinum compounds and anthracyclines, and also ionizing radiation induce oxidative stress generating high levels of ROS [92]. The induction of oxidative stress can lead to the preferential killing of cancer cells. Currently, the main problem of chemotherapy and radiotherapy is the development of resistance mechanisms; recent works report the role of miRNAs in the response to these therapies by targeting oxidative stress molecules. Recently, it was demonstrated that miR-125b is involved in chemotherapy resistance by affecting oxidative stress pathways in breast cancer [93]. MiR-125b, by targeting HAX-1, an anti-apoptotic gene, impacts on doxorubicin resistance. The mechanism behind this phenomenon is a decrease in the levels of MMP following HAX-1 downregulation and the release of ROS from the mitochondria into the cytoplasm. Thus, miR-125b is able to re-sensitize breast cancer cells to doxorubicin treatment using ROS pathway (Figure 3C). Concerning chemoresistance, Roscigno G. et al. have reported that miR-24, up-regulated in breast cancer stem cells, induces resistance to cisplatin by targeting the pro-apoptotic factor BimL [94]. Furthermore, miR-24 targets FIH1 that induces the repression of HIF-1α. Thus, the authors have shown that miR-24 is induced in hypoxic conditions, leading to cancer stem cell growth and consequently inducing chemotherapy resistance. Breast cancer patients often poorly respond to radiotherapy, and the mechanisms of radioresistance have not been elucidated yet. MiR-668 was found increased in breast cancer cells resistant to radiotherapy; this phenomenon occurs because IκBα is a direct target of miR-668, leading to the activation of NF-κB [95]. Generally, drug resistance is an important challenge in the treatment of breast cancer, especially for TNBC, which still don’t have target therapy. To date, novel therapeutic strategies have been tested mainly in the treatment of TNBC. MiR-223 is related to resistance to TRAIL-induced apoptosis in cancer stem cells of TNBC [96]. Indeed, reintroduction of miR-223 and treatment with TRAIL in MDA-MB-231 cell line induces a strong generation of ROS, through the targeting of HAX-1 into the mitochondria, and TNBC stem cells are more sensitive to TRAIL treatment. Moreover, the miR-223/HAX-1 axis enhances the sensitivity to doxorubicin and cisplatin in TNBC stem cells (Figure 3C).

## 7. Conclusive Remarks

In this review, we have illustrated what emerges from the literature about the important role of oxidative stress in the pathogenesis of breast cancer, influencing most of the pathways usually altered in tumors, affecting also response to therapy. Moreover, many of the proteins involved in this process, such as SOD2 and NRF2, can exert opposite roles depending on the context, complicating the scenario. Thus, it is important to explore in more detail the mechanisms behind the regulation of the redox status in relation to a specific scenario in order to better define which pathways can be proposed as therapeutic targets. MiRNAs act as regulative elements in almost every biological process, including oxidative stress and cancer. Here, we have mainly reviewed the literature concerning miRNAs involved in the regulation of oxidative stress players in breast cancer disease. MiRNA role in the regulation of redox status makes them as hypothetical and crucial targets or tools for therapy since they could provide the treatment context specificity. MiRNA general use as therapy option has yet to show relevant results, but an increasing body of evidence has been provided through the years in favor of such a solution, especially in oncology. Additionally, breast cancer is one of the most studied neoplasia and many are the miRNAs which mechanism of action is consolidated in this framework. Hopefully, therefore, it will be soon possible to have major improvements in this research field.

## Figures and Tables

**Figure 1 ijms-20-05143-f001:**
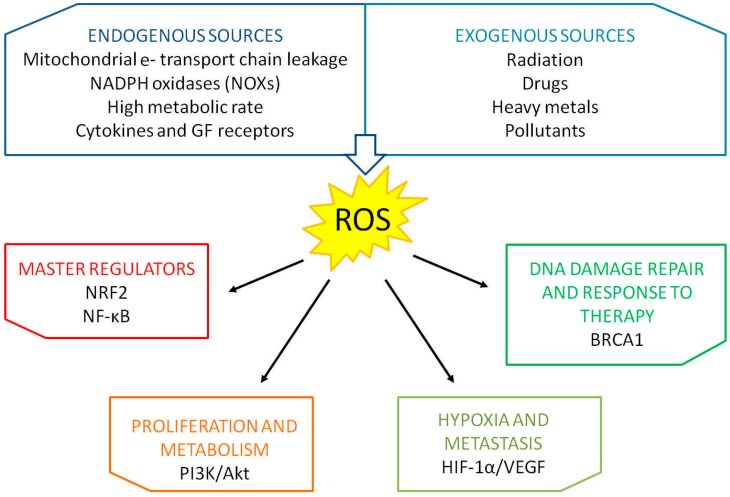
Endogenous and exogenous sources of ROS and pathways influenced by oxidative stress in breast cancer.

**Figure 2 ijms-20-05143-f002:**
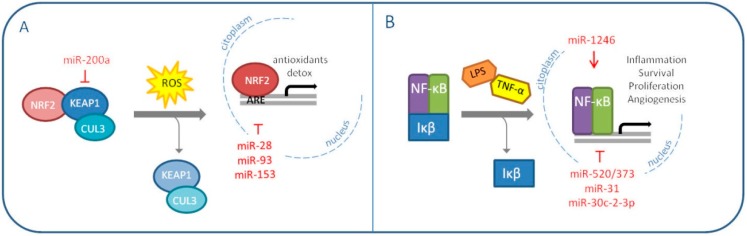
MiRNAs modulating oxidative stress master regulators NRF2 (**A**) and NF-κB (**B**) in breast cancer (The red arrow indicates upmodulation, the red “T” stands for inhibition).

**Figure 3 ijms-20-05143-f003:**
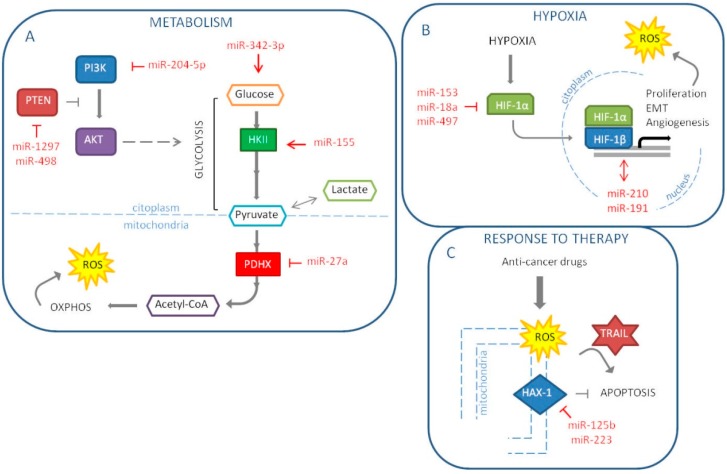
MiRNAs involved in the regulation of hallmarks of cancer influenced by oxidative stress in breast cancer: metabolism (**A**), hypoxia (**B**) and response to therapy (**C**) (The red arrow indicates upmodulation, the red “T” stands for inhibition).

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
