# Peer review of "MicroRNA and Oxidative Stress Interplay in the Context of Breast Cancer Pathogenesis"

_ijms, 2019, doi:10.3390/ijms20205143_

Round 1
Reviewer 1 Report
In the following review authors summarized the correspondence between miRNAs and oxidative stress in breast cancer tumorigenesis. I have two minor comments:
Is there any data concerning expression of the described miRNAs and oxidative stress between initiation of tumorigeneis versus tumor progression?Do the miRNAs expression and ROS significantly changes throughout disease duration? I suggest to summarize results in the table, it will be more readable.
Author Response
We thank the reviewer for the interesting questions. We searched the literature to provide the additional information requested but there is still no data describing differential expression of the cited miRNAs between early and late stage of breast cancer in relation to ROS levels. We do not exclude that there could be evidence concerning other miRNAs or other tumor types. ROS levels are high throughout the whole process of tumor development due to sustained proliferation, but cancer cells also increase the expression of antioxidants to avoid lethal damages. Chemotherapeutic drugs disturb this “aberrant equilibrium” in favor of ROS production to trigger apoptosis. Consequently, chemoresistant cells activate opposing pathways to survive. An example of such mechanism, which involves also miRNAs, is the work by Mateescu et al, published on Nature Medicine in 2011. The authors showed that the ROS-induced upregulation of miR-141 and miR-200a, two members of the miR-200 family, promotes ovary tumorigenesis in mouse models. These miRNAs target p38α, a member of the MAPK stress response pathway. Upon treatment with paclitaxel, the miRNA-mediated downmodulation of p38α becomes counterproductive for the tumor. In fact, in resistant tumors, downregulation of the miRNAs directly boosts p38α and ZEB1 or ZEB2 activity, inducing an epithelial-mesenchymal transition–like phenotype, which favors metastasis dissemination. Therefore, the mechanism illustrated here attributes two opposite role to miR-141 and miR-200a in this specific scenario. The authors do not speculate whether the two miRNAs would have been downmodulated during tumor progression even in an untreated condition.
In conclusion, despite the intriguing issue, we did not find any new relevant information to add to our manuscript.
Mateescu B., Batista L., Cardon M., Gruosso T., de Feraudy Y., Mariani O., Nicolas A., Meyniel J.P., Cottu P., Sastre-Garau X., Mechta-Grigoriou F. miR-141 and miR-200a act on ovarian tumorigenesis by controlling oxidative stress response. Nat Med. 2011 Nov 20;17(12):1627-35.
Reviewer 2 Report
Cosentino et al. have done a fantastic review on microRNAs and oxidative stress in breast cancer. I have enjoyed reading this review, which exposes a nice overview of the field.
In general, the review paper is well constructed and acknowledgeable. Each section has distinct purpose and the topic is interesting.
This review is complete and the figures are well designed, however I have some minor comments below.
Other comments:
The authors should revise minor typos such:
- page 2, in the Figure1 NFkB (the key is a kappa),
- page 3, 4, 5 in the titles, Mirna instead of miRNA
Revise all the titles and subtitles. I would recommend to increase the resolution of the figures, they look pixelated.
Author Response
We thank the reviewer for the enthusiastic reception of our work. As suggested, we revised figures, titles and subtitles and we have increased image resolution to 600 dpi.